# Assessment of the Undrained Shear Strength and Settlement of Organic Soils under Embankment Loading Using Artificial Neural Networks

**DOI:** 10.3390/ma16010125

**Published:** 2022-12-23

**Authors:** Zbigniew Lechowicz, Maria Jolanta Sulewska

**Affiliations:** 1Department of Geotechnical Engineering, Institute of Civil Engineering, Warsaw University of Life Sciences—SGGW, Nowoursynowska 159 St., 02-776 Warsaw, Poland; 2Faculty of Civil and Environmental Engineering, Bialystok University of Technology, Wiejska 45E St., 15-351 Bialystok, Poland

**Keywords:** organic soils, embankment design, settlement, undrained shear strength, statistical analysis, artificial neural networks

## Abstract

In engineering practice, due to the high compressibility and very low shear strength of organic soils, it is difficult to construct an embankment on organic subsoil. High variability and significant change in geotechnical parameters cause difficulties in predicting the behavior of organic soils under embankment loading. The aim of the paper was to develop empirical relationships used in the preliminary design for evaluating the settlement and undrained shear strength of organic subsoil loaded by embankment based on data obtained from four test sites. Statistical multiple regression models were developed for evaluating the settlement in time and undrained shear strength in time individually for peat and gyttja. Neural networks to predict the settlement and undrained shear strength in time for peat and gyttja simultaneously as double-layer subsoils as well as a separate neural network for peat and a separate neural network for gyttja as single-layer subsoils were also developed. The vertical stress, thickness, water content, initial undrained shear strength of peat and gyttja, and time were used as the independent variables. Artificial neural networks are characterized by greater prediction accuracy than statistical multiple regression models. Multiple regression models predict dependent variables with maximum relative errors of about 35% to about 60%, and neural networks predict output variables with maximum relative errors of about 25% to about 30%.

## 1. Introduction

A general problem in many countries, including Poland, is that many geotechnical structures such as dam embankments, levees, and dikes, and road embankments have to be often built on organic subsoil [1,2,3]. In engineering practice, due to the high compressibility and very low initial undrained shear strength of organic soils, it is difficult to construct an embankment even of a relatively low height. For the design of embankments on organic subsoil, important problems which have to be solved are the assessment of embankment stability and subsoil deformations. Stability analysis has to be carried out for different loading conditions but an evaluation of the stability during construction is of major importance. The stability analysis of embankment during construction is usually based on undrained shear strength *τ_fu_* of organic soils. For a stage-constructed embankment, time and money can be saved if the undrained shear strength increase in the organic soils and their settlement beneath the embankment can be accurately predicted [4].

The behavior of soft clays and organic soils has been studied and analyzed by many researchers in terms of primary and secondary compression [5], isotaches concept [6,7], rate process [8], resistance concept [9], and creep deformations [10,11,12,13]. The high complexity of the behavior of Holocene organic soils under loading results primarily from their large compressibility with creep effects, low undrained shear strength, significant changes in permeability with porosity changes, non-linear material characteristics, and significant spatial variability of properties [1,14,15]. In the preliminary design of embankments on organic subsoil, the prediction of settlement and the change in undrained shear strength can be carried out based on empirical relationships often obtained from the statistical analysis [16,17,18,19,20,21,22,23,24,25]. The final settlement of organic soils can be estimated using empirical relationships based on the thickness of organic soils, water content or void ratio of organic soils, and stress increase [1,2]. The final increase in undrained shear strength of organic soils can be evaluated using empirical relationships as a function of effective stress increase [14,18]. There is a lack of empirical relationships to predict both the settlement and undrained shear strength of organic soils in time. For the final design, various theories [26,27,28] and numerical models [29,30,31,32,33,34,35,36] taking into account to varying degrees the above-mentioned factors to predict the deformations and the change in undrained shear strength are used. These methods require the determination of the soil characteristics and parameters carried out based on field and laboratory tests [37,38,39].

Statistical procedures (linear and non-linear, single and multiple regression models) [40] and more and more often—artificial neural networks (ANNs) are most often used as tools for the analysis of the results of empirical studies of geotechnical parameters [41,42,43]. Neural networks are analytical techniques modeled on the simplified learning process of the neurological functions of the brain of living beings. They are able to predict new observations (variables) from other observations (same or different variables) after learning from the existing data [44,45]. Among the many types and topologies of neural networks, the multi-feed-forward layered neural networks of the Multi-Layer Perceptron (MLP) type are most often used in regression problems [46,47]. They have neurons organized into layers. The input layer introduces to the network *i* input variables (*Xi*). The output layer neurons calculate the output variables *j* of the entire network (*Yj*). The neurons of the hidden layers and the output layer are connected to all the neurons of the previous layer. Neural networks perform calculations based on the principle of parallel operation of a system of individual neurons, each of which acts as a specific signal converter. Each neuron calculates its activation level by taking the weighted sum of the neuron outputs of the previous layer, then transforming this sum with the activation function to obtain its output variable. The network unknowns, i.e., the number of network parameters (*NNP*) are weights and threshold values (biases) of individual neurons. The process of learning the network is aimed at such a correction of the set of weights and biases to obtain the smallest value of the error function, i.e., the smallest difference between the actual value of the output variable and the approximated value. Artificial neural networks are non-linear tools taught by iterative algorithms. Artificial neural networks can be used to predict both the settlement and undrained shear strength of organic soils in time under embankment loading.

Shahin et al. [48] presented the state-of-the-art examination of artificial neural networks in geotechnical engineering. The use of artificial neural networks to predict the settlement of shallow foundations on granular soils has been shown by Shahin et al. [49] and to predict the settlement of a stone column under a highway embankment by Chik et al. [50].

The aim of the paper was to develop empirical relationships used in the preliminary design for evaluating the settlement and undrained shear strength in the time of organic subsoil loaded by embankment based on data obtained from four test sites. Statistical multiple regression models and neural network models were developed for evaluating the settlement in time and undrained shear strength in time for peat and gyttja. Applying the design and build system the artificial neural networks supplemented by new data can be also used during embankment construction. This paper presents the results of field and laboratory tests of organic soils at the Białośliwie, Antoniny, Wonieść, and Koszyce test sites. Organic subsoils consist peat layer (amorphous and moderately decomposed) and calcareous soil called “gyttja” (calcareous and calcareous-organic) loaded by test embankments. The tested amorphous and moderately decomposed peats were with very high porosity being the ratio of pore volume to total volume *n* = 0.80–0.88 and organic content *I_om_* between 65% and 90%. The calcareous gyttja and calcareous-organic gyttja were with high porosity *n* = 0.70–0.75, calcium carbonate content CaCO_3_ between 65% and 90%, and organic content *I_om_* between 6% and 30%. Due to very high water content, the calcium carbonate and organic content matters are very soft with non-linear material characteristics, creep effect under loading, and significant spatial variability of properties. Data from four test sites concerning the settlement and undrained shear strength measurements were collected to elaborate a method for evaluating the settlement and undrained shear strength of organic soils in time based on statistical analysis. However, the statistical regression models may sometimes show relatively high values of maximum relative error. Therefore, a method for evaluating the settlement and undrained shear strength of organic subsoil loaded by embankment in time using artificial neural networks based on data obtained from four test sites is presented in this study. The presented neural network, with an architecture of 9–5–4, predicts the settlement in time and undrained shear strength in time simultaneously for peat and gyttja as double-layer subsoil based on four independent variables: vertical stress in peat and gyttja, the thickness of peat and gyttja layers, the water content of peat and gyttja, initial undrained shear strength of peat and gyttja, and a common independent variable: time. A separate neural network for peat with an architecture of 5–5–2 and a separate neural network for gyttja 5–5–2 as single-layer subsoils were also developed.

## 2. Materials and Methods

### 2.1. Characteristics of the Test Sites

#### 2.1.1. Białośliwie Site

The Białośliwie test site is located in north-western Poland in the Noteć River valley where the Department of Geotechnical Engineering of the Warsaw University of Life Sciences performed extensive field and laboratory investigations during the construction of pound dikes in the 1980-ties [51]. Four test embankments (with and without vertical prefabricated drains) on peat-gyttja subsoil and on peat subsoil were constructed in one stage. The embankment without vertical drains No. 2 with a height of 2 m caused vertical stress of 35 kPa on the peat layer and 32 kPa on the gyttja layer and embankment No. 4 with a height of 2.2 m causing vertical stress of 40 kPa on peat subsoil (Figure 1) is considered in the paper.

#### 2.1.2. Antoniny Site

The Antoniny test site is located in north-western Poland in the Noteć River valley where the Department of Geotechnical Engineering of the WULS in cooperation with the Swedish Geotechnical Institute performed extensive field and laboratory investigations during the construction of river levees [52,53,54,55]. Two test embankments (with and without vertical prefabricated drains) were constructed in three stages between 1983–1987. The embankment without vertical drains constructed in three stages (Figure 2) was then brought to failure by successively increasing the height of the fill [53]. In the first stage, the embankment height was 1.2 m, in the second stage, 2.5 m, and in the third stage, 3.9 m. The increase in the embankment height in the following stages caused the increment of vertical stress 21 kPa, 20 kPa, and 19 kPa on the peat layer and 20 kPa, 19 kPa, and 18 kPa on the gyttja layer, respectively.

#### 2.1.3. Wonieść Site

The Wonieść test site is located in western Poland in the Samica River valley where the Department of Geotechnical Engineering of the WULS performed extensive field and laboratory investigations during the construction of embankment dams in the 1980-ties [56]. The main dam embankment was constructed on peat-gyttja subsoil in one stage on a reinforcement steel mattress (Figure 3). The embankment height was 4 m causing vertical stress of 70 kPa on the peat layer and 60 kPa on the gyttja layer.

#### 2.1.4. Koszyce Site

The Koszyce test site is located in north-western Poland in the Ruda River valley where a laboratory and field testing program was performed below and outside of the main dam embankment of the Koszyce reservoir (Figure 4) [57]. Due to the presence of deep soft organic subsoil, the dam embankment was constructed on a reinforcement steel mattress. Very low undrained shear strength of the organic soils with significant variability in thickness combined with relatively quick embankment loading caused a break of the mattress and embankment failure during the construction with very large deformations of organic subsoil in the central part of the Koszyce dam [57]. The cross-sections of the embankment dam outside of the failure zone for a period of three years of construction and eleven years of maintenance are considered in the paper. The embankment heights at the three selected profiles were 6 m causing vertical stress of 100 kPa on the peat layer and 95 kPa on the gyttja layer, 2 m causing vertical stress of 35 kPa on the peat layer, and 32 kPa on the gyttja layer and 4 m causing vertical stress 65 kPa on peat layer and 60 kPa on the gyttja layer.

### 2.2. Characteristics of the Tested Organic Soils

#### 2.2.1. Białośliwie Site

At the Białośliwie site, the virgin subsoil consists of a sedge peat layer and a calcareous-organic soil layer called “gyttja” underlain by a fine sand layer. Based on origin as well as organic content *I_om_* and calcium carbonate content CaCO_3_, peat and gyttja were classified [1]. Under embankment No. 2, the organic subsoil with a thickness of 4.0 m, consists of a 2.0 m thick peat layer and a 2.0 m gyttja layer. Under embankment No. 4, the organic subsoil consists of a 4.0 m thick peat layer. The peat layer is moderately decomposed, with water content *w_n_* between 410% and 430%, and organic content *I_om_* between 85% and 90%. The calcareous-organic gyttja layer is characterised by water content *w_n_* between 120% and 130%, organic content *I_om_* between 20% and 25%, and calcium carbonate content CaCO_3_ between 65% and 70%. The organic matter content *I_om_* was determined by combustion at a temperature of +550 °C [58]. The calcium carbonate content CaCO_3_ was determined by the gasometer method [58].

#### 2.2.2. Antoniny Site

At the Antoniny site, the virgin subsoil consists of an amorphous peat layer and a calcareous-organic gyttja, and calcareous gyttja underlain by a fine sand layer. The organic subsoil, 7.8 m thick, consisting 3.1 m thick peat layer and a 4.7 m gyttja layer. Organic soils are overconsolidated with an over-consolidation ratio (OCR), decreasing from 5 to 2 with depth. The amorphous peat layer is with water content *w_n_* between 310% and 340%, and organic content *I_om_* between 65% and 75%. The calcareous-organic and calcareous gyttja layers are characterised by water content *w_n_* between 110% and 140%, organic content *I_om_* between 8% and 20%, and calcium carbonate content CaCO_3_ between 70% and 90%.

#### 2.2.3. Wonieść Site

At Wonieść site the virgin subsoil consists of a sedge-reed peat layer and a calcareous-organic gyttja underlain by a fine sand layer. The organic subsoil, 11.3 m thick, consists of a 4.2 m thick peat layer and a 7.1 m gyttja layer. The moderately decomposed peat layer is with water content *w_n_* between 500% and 550%, and organic content *I_om_* between 65% and 80%. The calcareous-organic and calcareous gyttja layers are characterised by water content *w_n_* between 150% and 160%, organic content *I_om_* between 15% and 30%, and calcium carbonate content CaCO_3_ between 65% and 80%.

#### 2.2.4. Koszyce Site

At the Koszyce site, the central part of the dam embankment is founded on organic subsoil with a variable thickness. The soft subsoil consists of organic sediments with a thickness from 4 m to 27 m. The uppermost 2.5 m consists of peat. Below there is a layer of gyttja with a thickness from 2 m to 25 m. Below occurs a sand layer. The amorphous peat layer is characterised by water content *w_n_* between 500% and 550%, and organic content *I_om_* between 80% and 85%. The calcareous-organic and calcareous gyttja layers are characterised by water content *w_n_* between 110% and 140%, organic content *I_om_* between 6% and 20%, and calcium carbonate content CaCO_3_ between 65% and 80%.

### 2.3. Settlement and Undrained Shear Strength Measurements

The field investigations performed at the test sites included, among others, settlement measurements and evaluation of undrained shear strength from Field Vane Tests (FVT). In order to evaluate the undrained shear strength *τ_fu_* of organic soils from FVT, the measured shear strength *τ_fv_* was corrected by using the correction factor *μ* according to the formula:(1)$$\tau_{fu}=\mu \cdot \tau_{fv},$$

The correction factor *μ* was evaluated according to the method elaborated by the Swedish Geotechnical Institute (SGI) [59]. Other methods used to evaluate the correction factor μ were presented in the literature [1,52,53].

#### 2.3.1. Białośliwie Site

The settlement measurements of peat and gyttja at the Białośliwie site were carried out on the basis of levelling the surface settlement gauge (installed 0.5 m below ground level, at the peat surface) and deep settlement gauge (installed at the gyttja surface) for the test embankment No. 2 and the surface settlement gauge (installed 0.5 m below at the peat surface) for the test embankment No. 4. Therefore, for settlement analysis the initial thickness of peat and of gyttja layers were 1.5 m and 2.0 m for test embankment No. 2 and of peat layer was 3.5 m for test embankment No. 4, respectively. In order to evaluate the undrained shear strength *τ_fu_* of organic soils from FVT, the measured shear strength *τ_fv_* was corrected by using the correction factor *μ* according to the SGI method for peat *μ* = 0.5 and for gyttja *μ* = 0.7. The settlements and undrained shear strength versus the time of peat and gyttja at the Białośliwie site for the test embankment No. 2 and No. 4 are presented in Figure 5 and Figure 6.

#### 2.3.2. Antoniny Site

The settlement measurements of peat and gyttja at the Antoniny site for three stages were carried out on the base of levelling the surface settlement gauge (installed 0.5 m below ground level, at the peat surface) and deep settlement gauge (installed at the gyttja surface). Taking into account individually each stage, for settlement analysis, the initial thickness of peat and of gyttja layers for the first stage were 2.60 m and 4.70 m, for the second stage 2.36 m and 4.60 m and for the third stage 2.08 m and 4.16 m, respectively. The correction factor *μ* according to the SGI method was used for peat *μ* = 0.5 and for gyttja *μ* = 0.7–0.8. The settlements and undrained shear strength versus the time of peat and gyttja at the Antoniny site for three stages are presented in Figure 7.

#### 2.3.3. Wonieść Site

The settlement measurements of peat and gyttja at the Wonieść site were carried out on the base of levelling the surface settlement gauge (installed 0.5 m below the peat surface) and deep settlement gauge (installed at the gyttja surface). Therefore, for settlement analysis the initial thickness of peat and of gyttja layers were 3.7 m and 7.1 m, respectively. The correction factor μ according to the SGI method was used for peat *μ* = 0.5 and for gyttja *μ* = 0.6. The settlements and undrained shear strength versus the time of peat and gyttja at the Wonieść site are presented in Figure 8.

#### 2.3.4. Koszyce Site

In order to evaluate the change in the geometry of the organic subsoil at the Koszyce site during the period of three years of construction and eleven years of maintenance, the field investigation including borings, soundings, and settlement measurements has been analysed. Therefore, for settlement analysis, three different heights of the dam embankment were selected with the initial thickness of peat layer 2.0 m and of gyttja layers 14.0 m and 12.0 m, respectively. The correction factor μ according to the SGI method was used for peat *μ* = 0.5 and for gyttja *μ* = 0.6–0.7. The settlements and undrained shear strength versus the time of peat and gyttja at the Koszyce site for three profiles are presented in Figure 9.

Figure 5, Figure 6, Figure 7, Figure 8 and Figure 9 show the measurement results of the settlement and undrained shear strength of organic soils (peat and gyttja) during the consolidation process caused by the embankment loading. The presented measurement results were used as dependent variables to develop regression and neural models. In engineering practice, there is a need to develop empirical relationships using easy-to-determine parameters for the preliminary design of embankments on organic subsoil to the prediction of settlement and undrained shear strength in time. Based on the analysis of many parameters, the vertical stress, thickness, water content, initial undrained shear strength of peat and gyttja, and time were initially selected as independent variables.

## 3. Results

The database for the analysis consisting 82 cases described by a total of 13 variables: nine independent variables *X*1–*X*9 and four dependent variables *Y*1–*Y*4. Notations of independent and dependent variables and the statistical specifications of the database used in the regression models are presented in Table 1.

A detailed summary of the test results of the analyzed variables is presented in Table 2.

It can be seen that the tests were carried out in a wide range of time *t*, vertical stress *σ*, and for a large range of thickness of the gyttja layer.

## 4. Analysis of Test Results

Statistical and neural calculations were performed using the STATISTICA version 12 computer program [60]. Statistical analyzes were performed assuming the significance level *α* = 0.05.

### 4.1. Statistical Analysis of Results

In the beginning, the importance and significance of individual independent variables and their correlation with dependent variables were estimated. On the basis of the coefficient matrix of a simple linear correlation *r*, it was found that for peat and for gyttja, there are statistically significant correlations between the dependent variables *Yj* and the independent variables *Xi*. Matrices of simple linear correlation coefficients *r* for peat and gyttja are presented in Table 3 and Table 4. Statistically significant correlation coefficients *r* are marked in bold. The remaining simple correlations between the variables are statistically insignificant.

For peat, the coefficients *r* of linear simple correlation of individual regression equations are: for *Y*1 = f (*Xi*) where *i* = 1, 3, 5, 7, 8, *r* = (−0.288)–0.617 (with the most influential variable being *X*1 when *r* = 0.617) and for *Y*3 = f (*Xi*) where *i* = 1, 5, 7, 8, except *X*3 when *r* = −0.165, *r* = 0.331–0.683 (with the most influential variable being *X*7 when *r* = 0.683).

For gyttja, the coefficients *r* of linear simple correlation of individual regression equations are *Y*2 = f (*Xi*) where *i* = 2, 4, 6, 7, 9, *r* = 0.369–0.821 (with the most influential variable being *X*7 when *r* = 0.821), and *Y*4 = f (*Xi*) where *i* = 2, 4, 6, 7, 9, *r* = 0.639–0.815 (with *X*9 being the most influential variable when *r* = 0.815).

Linear multiple regression models for peat were developed: *Y*1 = f (*X*1, *X*3, *X*5, *X*7, *X*8) and *Y*3 = f (*X*1, *X*5, *X*7, *X*8). Linear multiple regression models for gyttja were developed: *Y*2 = f (*X*2, *X*4, *X*6, *X*7, *X*9) and *Y*4 = f (*X*2, *X*4, *X*6, *X*7, *X*9).

Standard estimation errors *SEE* were calculated based on the obtained regression equations, using the Formula (2):(2)$$SEE=\sqrt{\frac{\sum_{1}^{n}{\left(y^{\left(p\right)}-d^{\left(p\right)}\right)}^{2}}{n-2}}$$

The standard estimation errors *SEE* informs about the average value of empirical deviations of the value of the dependent variable from the value of the dependent variable calculated from the model, i.e., informs about the degree of fit of the model to the empirical data [57]. The smaller the *SEE*, the better the model fits.

Relative prediction errors *RE* were calculated based on the obtained regression equations, using the Formula (3):(3)$$RE=\left|\frac{y^{\left(p\right)}-d^{\left(p\right)}}{y^{\left(p\right)}}\right|\cdot 100\%,$$
where *y*^(*p*)^ is the value obtained from the tests, *d*^(*p*)^ is the value obtained by the model, *p* = 1, …, *n*, where *n* is the number of patterns for a given variable in the analyzed data set.

The obtained regression equations and their coefficients of determination *R*^2^, the standard estimation errors ±*SEE*, and the maximum relative prediction errors ±*REmax* are presented in Table 5. The models included only statistically significant variables, and statistically insignificant variables were omitted.

It can be seen, the maximum relative prediction error REmax using the multiple regression equations of peat and gyttja settlement is about 60%, and the relative error of predicting undrained shear strength is 40% and 35%, respectively.

### 4.2. Analysis of Results Using Artificial Neural Networks

To solve the regression problems, one-way networks Multi-Layer Perceptron (*MLP*) type with the *N*-*H*-*M* architecture were used, where *N* is the number of input variables, *H* is the number of neurons in one hidden layer and *M* is the number of output variables.

Three neural networks were developed:9–5–4 network for simultaneous prediction of four dependent variables Y1-Y4 for a subsoil consisting of two layers—peat and gyttja,5–5–2 network for simultaneous prediction of two dependent variables Y1 and Y3 when the subsoil is single-layer—only from peat,5–5–2 network for simultaneous prediction of two dependent variables Y2 and Y4 when the subsoil is single-layered—only from gyttja.

Neural networks with four outputs were used, calculated simultaneously for peat and for gyttja (*Y*1 and *Y*3 as well as *Y*2 and *Y*4). The entire data set with the total cases number of *n* = 82 was randomly divided into subsets: the training subset *L* = 66 cases, the validation subset *V* = 8 cases, and the testing subset *T* = 8 cases. The training subset *L* used to train the network, the validation subset *V* used to determine the number of network learning epochs—“early stopping”, i.e., the number of epochs at which the error in the validation set is the smallest [46]. The testing subset *T* was not used for training and validation and was used for the final one-time assessment of the predictive quality neural network. Multiple neural networks were analyzed with the same case selection for subsets *L*, *V*, and *T*.

A neural network with the 9–5–4 architecture was selected, i.e., a network with nine inputs (*X*1-*X*9), four neurons in the hidden layer, and four outputs (*Y*1–*Y*4). An exponential function was chosen from among several activation functions (linear, logistic, tanh, exponential) in hidden neurons, and a linear function in output neurons. The neural network was trained by the Variable Metric Method with the Broyden-Fletcher-Goldfarb-Shanno (BFGS) algorithm [46,60] for 94 epochs (i.e., learning cycles for all cases of the *L* subset). The number of network parameters (i.e., the number of neuron connection weights and biases) was *NNP* = *N* · *H* + *H* · *M* + *H* + *M* = 9 · 4 + 4 · 4 + 4 + 4 = 60 < *L* = 66.

Separate neural networks were also developed: for peat, the 5–5–2 BFGS for 120 epochs network with five inputs (*X*1, *X*3, *X*5, *X*7, *X*8), five hidden neurons, and two outputs (Y1, *Y*3) and for gyttja, the network 5–5–2 BFGS for 91 epochs with five inputs (*X*2, *X*4, *X*6, *X*7, *X*9), five hidden neurons, and two outputs (*Y*2, *Y*4). For both networks, the activation functions of hidden neurons were exponential, and for output neurons, tangensoidal tanh. All cases were randomly divided into subsets with the following numbers: *L* = 58, *V* = 12, and *T* = 12. The number of unknown networks parameters was *NNP* = 5 · 5 + 5 · 2 + 5 + 2 = 42 < *L* = 58.

The obtained artificial neural networks, *R*^2^ determination coefficients, and *REmax* errors in the prediction of individual dependent variables are presented in Table 6.

The *RE* errors were calculated according to the Formula (3). The value of the error function of the entire neural network was calculated as a Sum of Squares (SOS) [60] according to the Formula (8):(8)$$SOS=\;\frac{1}{n}\overset{n}{\underset{p=1}{\sum }}\overset{M}{\underset{j=1}{\sum }}{\left(y_{j}^{\left(p\right)}-d_{j}^{\left(p\right)}\right)}^{2},$$
where *p* is the number of patterns in the analyzed subset, *j* is the number of output neurons.

**Table 6 materials-16-00125-t006:** ANNs with the best prediction accuracy and its error measures.

Output-Parameter	Architecture of ANN	*R* ^2^			*±REmax* in Set (*L + V + T*) (%)	Formula Number
		*L*	*V*	*T*		
*Y*1 *sp*	9–5–4 (BFGS 94)	0.914	0.910	0.943	25	(9)
*Y*2 *sg*		0.960	0.984	0.976	25	
*Y*3 *τp*		0.960	0.982	0.986	25	
*Y*4 *τg*		0.954	0.982	0.970	25	
*Y*1 *sp*	5–5–2 (BFGS 120)	0.946	0.856	0.931	30	(10)
*Y*3 *τp*		0.952	0.944	0.910	25	
*Y*2 *sg*	5–5–2(BFGS 91)	0.984	0.954	0.949	30	(11)
*Y*4 *τg*		0.980	0.978	0.982	25	

In addition, a global sensitivity analysis for neural networks [60] was carried out, which showed that for the 5–5–2 neural network for peat, the most influential are the variables (arranged in descending order): *X*7, *X*8, *X*3, *X*5, and *X*1, and for the 5–5–2 neural network for gyttja the variables: *X*6, *X*7, *X*9, *X*4, and *X*2.

The prediction accuracy, i.e., the maximum relative error of the *REmax* prediction of the individual dependent variables, is shown in Figure 10, Figure 11 and Figure 12.

The neural network with the 9–5–4 architecture simultaneously predicted the values of four output variables for the subsoil consisting of layers of peat and gyttja (*Y*1–*Y*4).

In the case of a single-layer subsoil, a separate 5–5–2 neural network was used, predicting two dependent variables for the peat layer (*Y*1 and *Y*3) and a separate 5–5–2 network for the gyttja layer (*Y*2 and *Y*4).

The maximum relative error of the *REmax* prediction of the 9–5–4 neural network, simultaneously predicting the values of the four output variables: *Y*1 *sp*, *Y*2 *sg*, *Y*3 *τp,* and *Y*4 *τg*, was about 25% for all dependent variables.

The maximum relative error of the *REmax* prediction of the 5–5–2 neural network, simultaneously predicting the values of the two output variables for the peat layer: *Y*1 *sp* and *Y*3 *τp*, was approximately 30% and 25%, respectively.

The maximum relative error of *REmax* prediction of the 5–5–2 neural network, simultaneously predicting the values of the two output variables for the gyttja layer: *Y*2 *sg* and *Y*4 *τg*, was approximately 30% and 25%, respectively.

## 5. Discussion

Based on the comparison of the prediction error measures, it can be seen that, in the studied range of data values, the multiple regression models have determination coefficients *R*^2^ = 0.632 and 0.881 (i.e., explain 63.2 and 88.1 of data variability) for the equations for predicting peat or gyttja settlement, respectively, and the maximum relative prediction errors of both equations are of approximately *REmax* = ±60%. The equations for predicting the undrained shear strength for peat and gyttja have *R*^2^ = 0.699 and 0.888 and *REmax* = ±40% and ±35%, respectively.

The analysis of obtained results indicates that the prediction of the settlement in time and undrained shear strength in time in the case of single-layer subsoils consisted peat or gyttja layers can be done using a separate neural network for peat with an architecture of 5–5–2 and a separate neural network for gyttja 5–5–2. In the case of double-layer of subsoil consisting peat and gyttja layers, the prediction of the settlement in time and undrained shear strength in time can be done using the presented neural network with an architecture of 9–5–4.

An artificial neural network with four outputs 9–5–4 has the error measures for output variables: coefficient of determination *R*^2^ = 0.943–0.986 in the testing subset, and for all output variables a maximum relative prediction errors, *REmax* = ±25%. Neural network 5–5–2 for peat has the errors measures for settlement and undrained shear strength respectively: *R*^2^ = 0.931 and 0.910 in the testing subset, and *REmax* = ±30% and 25%. Neural network 5–5–2 for gyttja has *R*^2^ = 0.949 and 0.982 in the testing subset, and *REmax* = ±30% and ±25%, respectively.

## 6. Conclusions

In this study, regression and neural modeling of settlement and undrained shear strength in the time of organic subsoil loaded by embankment was carried out. Empirical modeling was applied based on the results of field and laboratory tests of organic soils, using multiple regression equations and artificial neural networks of the MLP type. For the preliminary design of embankments on organic subsoil to the prediction of settlement and undrained shear strength in time the empirical relationships based on the independent variables: the vertical stress, thickness, water content, initial undrained shear strength of peat and gyttja, and time were developed.

The neural model presented in this study provided a more reliable prediction of the settlement in comparison to the statistical models, with a maximum relative prediction error *REmax* of peat about ±25% and 30% and gyttja about ±25% and 30% for double-layer and single-layer subsoils as well as of the undrained shear strength of peat about ±25% and gyttja about ±25% for both cases. The statistical multiple regression models had greater errors (*REmax* = ±35–60%) than the neural models.

Based on the analysis of the research results, it can be concluded that artificial neural networks can be used to predict settlement and undrained shear strength in time of the organic subsoil within the range of data under investigation. Artificial neural networks are characterized by better prediction accuracy than statistical regression models. Artificial neural networks with four outputs can be used to predict both settlement and undrained shear strength in time of double-layer subsoil from peat and gyttja layers, neural networks with two outputs can be used to predict both settlement and undrained shear strength in time of a single-layer subsoil from peat or gyttja.

## Figures and Tables

**Figure 1 materials-16-00125-f001:**
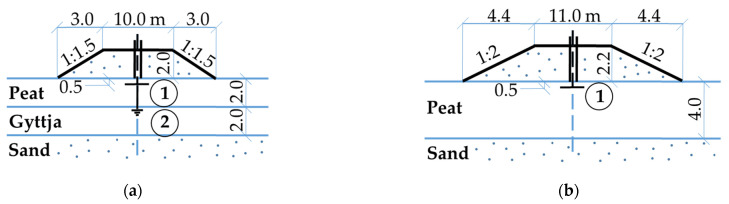
Test embankments at the Białośliwie site: (**a**) embankment No. 2, (**b**) embankment No. 4, 1—surface settlement gauge, 2—deep settlement gauge.

**Figure 2 materials-16-00125-f002:**
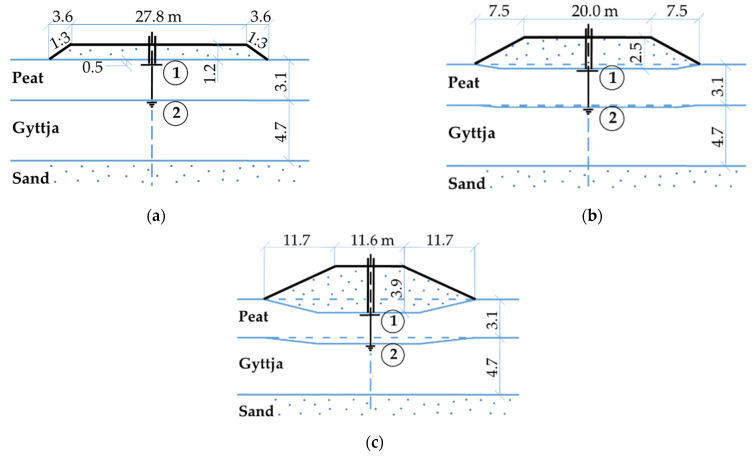
Test embankment at the Antoniny site: (**a**) 1st stage, (**b**) 2nd stage, (**c**) 3rd stage, 1—surface settlement gauge, and 2—deep settlement gauge.

**Figure 3 materials-16-00125-f003:**
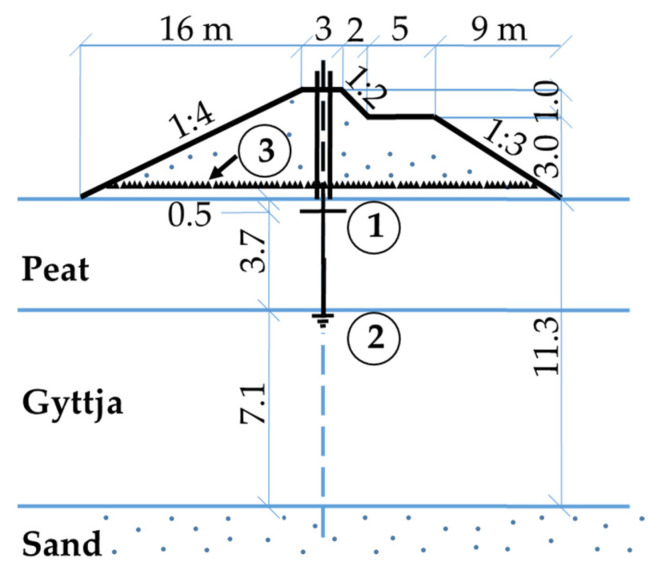
Test embankment at the Wonieść site: 1—surface settlement gauge, 2—deep settlement gauge. 3—reinforcement steel mattress.

**Figure 4 materials-16-00125-f004:**
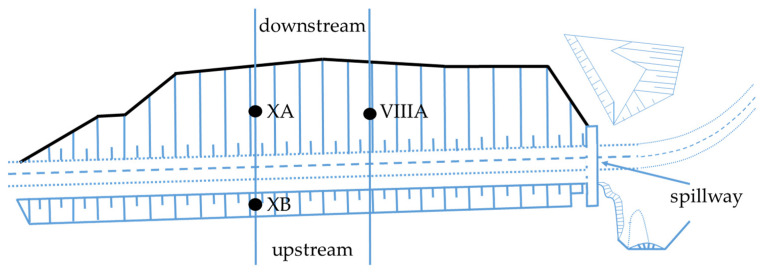
The embankment dam at the Koszyce site, • profiles.

**Figure 5 materials-16-00125-f005:**
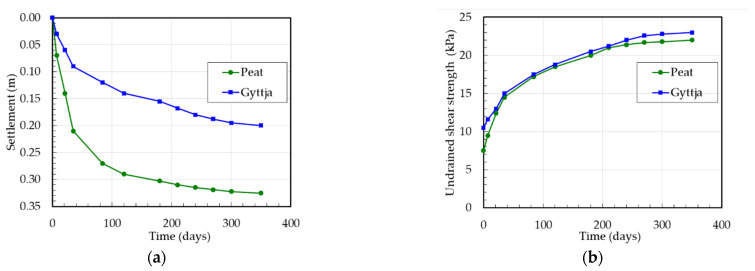
Test embankment No. 2 at the Białośliwie site: (**a**) settlement versus time, (**b**) undrained shear strength versus time.

**Figure 6 materials-16-00125-f006:**
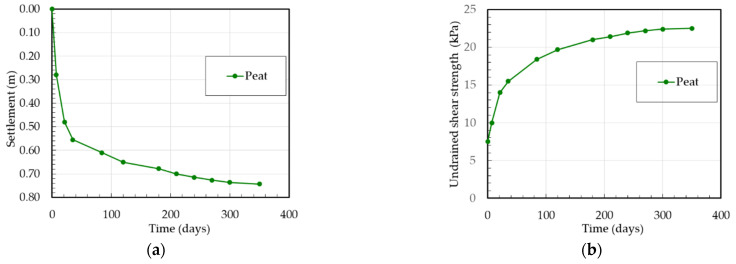
Test embankment No. 4 at the Białośliwie site: (**a**) settlement versus time, (**b**) undrained shear strength versus time.

**Figure 7 materials-16-00125-f007:**
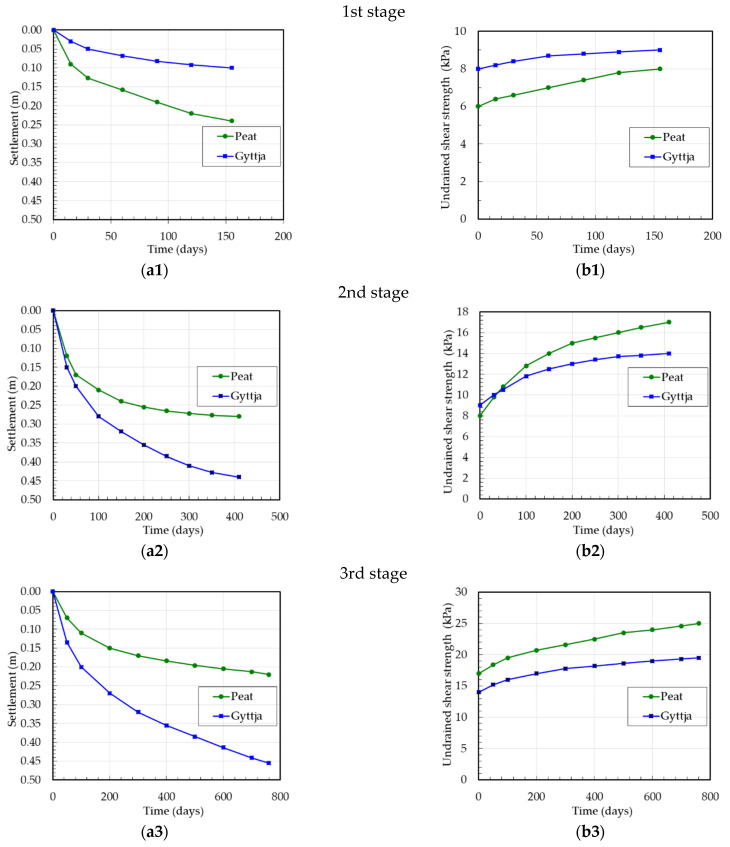
Test embankment at the Antoniny site: (**a1**–**a3**) settlement versus time, (**b1**–**b3**) undrained shear strength versus time.

**Figure 8 materials-16-00125-f008:**
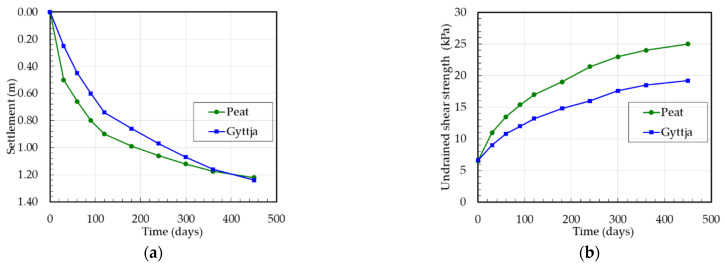
Test embankment at the Wonieść site: (**a**) settlement versus time, (**b**) undrained shear strength versus time.

**Figure 9 materials-16-00125-f009:**
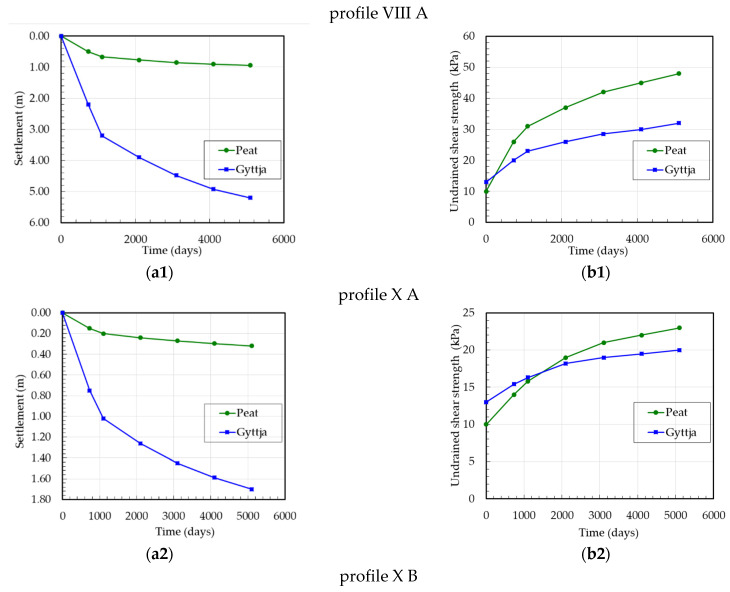
Test embankment at the Koszyce site: (**a1**–**a3**) settlement versus time, (**b1**–**b3**) undrained shear strength versus time.

**Figure 10 materials-16-00125-f010:**
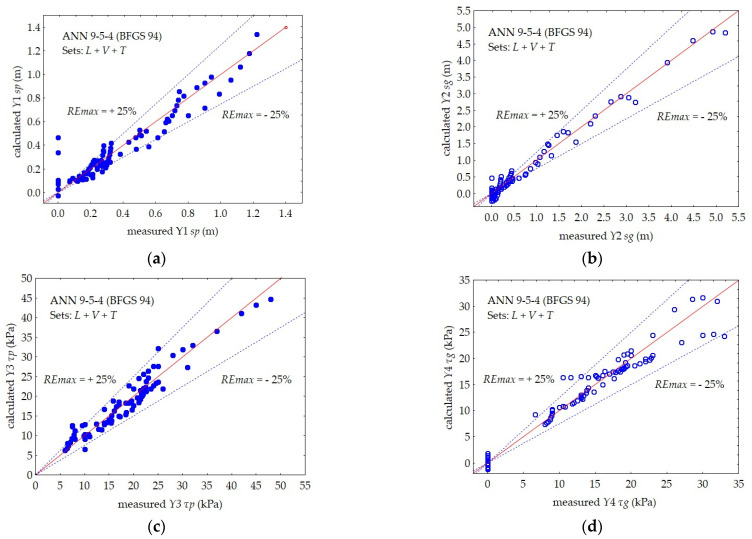
Comparison between the measured and calculated values by ANN 9–5–4 in subsets *L* + *V* + *T*: (**a**) *Y*1 *sp*, (**b**) *Y*2 *sg*, (**c**) *Y*3 *τp*, and (**d**) *Y*4 *τg*.

**Figure 11 materials-16-00125-f011:**
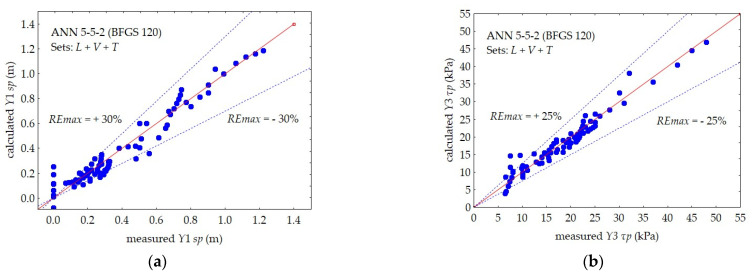
Comparison between the measured and calculated values by ANN 5–5–2 in subsets *L* + *V* + *T* for peat: (**a**) *Y*1 *sp*, (**b**) *Y*3 *τp*.

**Figure 12 materials-16-00125-f012:**
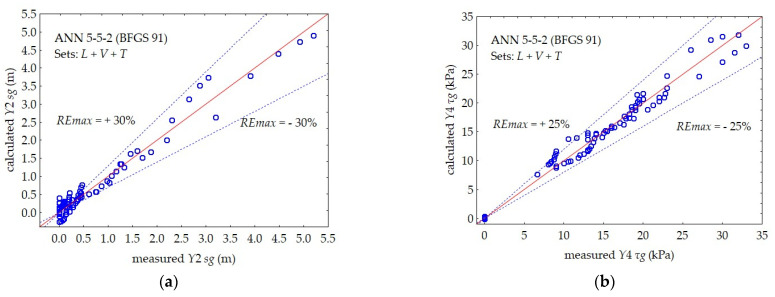
Comparison between the measured and calculated values by ANN 5–5–2 in subsets *L* + *V* + *T* for gyttja: (**a**) *Y*2 *sg*, (**b**) *Y*4 *τg*.

**Table 1 materials-16-00125-t001:** Variables analyzed and the statistical specifications of the database.

Variables	Notations	Minimum	Maximum	Average	Standard Deviation of Sample
*X*1 *σp*	Vertical stress on peat layer, (kPa)	19	100	43.1	24.6
*X*2 *σg*	Vertical stress on gyttja layer, (kPa)	0	95	34.2	26.7
*X*3 *hp*	Thickness of peat layer, (m)	1.5	3.7	2.5	0.8
*X*4 *hg*	Thickness of gyttja layer, (m)	0.0	14.0	6.2	5.0
*X*5 *wp*	Water content of peat, (%)	232	650	438.5	149.1
*X*6 *wg*	Water content of gyttja, (%)	0	150	102.6	45.8
*X*7 *t*	Time, (days)	0	5100	732.6	1294.4
*X*8 *τp*0	Initial undrained shear strength of peat, (kPa)	6.0	17.0	9.1	3.2
*X*9 *τg*0	Initial undrained shear strength of gyttja, (kPa)	0	14.0	9.2	4.5
*Y*1 *sp*	Settlement of peat layer, (m)	0.00	1.22	0.38	0.31
*Y*2 *sg*	Settlement of gyttja layer, (m)	0.00	5.20	0.74	1.17
*Y*3 *τp*	Undrained shear strength of peat, (kPa)	6.0	48.0	18.6	8.5
*Y*4 *τg*	Undrained shear strength of gyttja, (kPa)	0.0	33.0	14.4	8.4

**Table 2 materials-16-00125-t002:** Summary of input and output values for the statistical and ANN analyses.

No.	Site/ Variable	*X*1 σ*p* (kPa)	*X*2 σ*g* (kPa)	*X*3 *hp* (m)	*X*4 *hg* (m)	*X*5 *wp* (%)	*X*6 *wg* (%)	*X*7 *t* (days)	*X*8 *τp*0 (kPa)	*X*9 *τg*0 (kPa)	*Y*1 *sp* (m)	*Y*2 *sg* (m)	*Y*3 *τp* (kPa)	*Y*4 *τg* (kPa)
1	Białośliwie No. 2	35	32	1.50	2.00	410	120	0	7.5	10.5	0.000	0.000	7.5	10.5
2		35	32	1.50	2.00	410	120	7	7.5	10.5	0.070	0.030	9.5	11.6
3		35	32	1.50	2.00	410	120	21	7.5	10.5	0.140	0.060	12.4	13.0
4		35	32	1.50	2.00	410	120	35	7.5	10.5	0.210	0.090	14.5	15.0
5		35	32	1.50	2.00	410	120	84	7.5	10.5	0.270	0.120	17.2	17.5
6		35	32	1.50	2.00	410	120	120	7.5	10.5	0.290	0.140	18.5	18.8
7		35	32	1.50	2.00	410	120	180	7.5	10.5	0.303	0.155	20.0	20.5
8		35	32	1.50	2.00	410	120	210	7.5	10.5	0.310	0.168	21.0	21.2
9		35	32	1.50	2.00	410	120	240	7.5	10.5	0.315	0.180	21.4	22.0
10		35	32	1.50	2.00	410	120	270	7.5	10.5	0.319	0.188	21.7	22.6
11		35	32	1.50	2.00	410	120	300	7.5	10.5	0.323	0.195	21.8	22.8
12		35	32	1.50	2.00	410	120	350	7.5	10.5	0.326	0.200	22.0	23.0
13	Białośliwie No. 4	40	0	3.50	0.00	430	0	0	7.5	0.0	0.000	0.000	7.5	0.0
14		40	0	3.50	0.00	430	0	7	7.5	0.0	0.280	0.000	10.0	0.0
15		40	0	3.50	0.00	430	0	21	7.5	0.0	0.480	0.000	14.0	0.0
16		40	0	3.50	0.00	430	0	35	7.5	0.0	0.555	0.000	15.5	0.0
17		40	0	3.50	0.00	430	0	84	7.5	0.0	0.610	0.000	18.4	0.0
18		40	0	3.50	0.00	430	0	120	7.5	0.0	0.650	0.000	19.7	0.0
19		40	0	3.50	0.00	430	0	180	7.5	0.0	0.678	0.000	21.0	0.0
20		40	0	3.50	0.00	430	0	210	7.5	0.0	0.700	0.000	21.4	0.0
21		40	0	3.50	0.00	430	0	240	7.5	0.0	0.715	0.000	21.9	0.0
22		40	0	3.50	0.00	430	0	270	7.5	0.0	0.727	0.000	22.2	0.0
23		40	0	3.50	0.00	430	0	300	7.5	0.0	0.736	0.000	22.4	0.0
24		40	0	3.50	0.00	430	0	350	7.5	0.0	0.743	0.000	22.5	0.0
25	Antoniny No. 1/1	21	20	2.60	4.70	310	110	0	6.0	8.0	0.000	0.000	6.0	8.0
26		21	20	2.60	4.70	310	110	15	6.0	8.0	0.090	0.030	6.4	8.2
27		21	20	2.60	4.70	310	110	30	6.0	8.0	0.127	0.050	6.6	8.4
28		21	20	2.60	4.70	310	110	60	6.0	8.0	0.158	0.068	7.0	8.7
29		21	20	2.60	4.70	310	110	90	6.0	8.0	0.190	0.082	7.4	8.8
30		21	20	2.60	4.70	310	110	120	6.0	8.0	0.220	0.092	7.8	8.9
31		21	20	2.60	4.70	310	110	155	6.0	8.0	0.240	0.100	8.0	9.0
32	Antoniny No. 1/2	20	19	2.36	4.60	274	106	0	8.0	9.0	0.000	0.000	8.0	9.0
33		20	19	2.36	4.60	274	106	30	8.0	9.0	0.120	0.150	9.8	10.0
34		20	19	2.36	4.60	274	106	50	8.0	9.0	0.170	0.200	10.8	10.5
35		20	19	2.36	4.60	274	106	100	8.0	9.0	0.210	0.280	12.8	11.8
36		20	19	2.36	4.60	274	106	150	8.0	9.0	0.240	0.320	14.0	12.5
37		20	19	2.36	4.60	274	106	200	8.0	9.0	0.255	0.355	15.0	13.0
38		20	19	2.36	4.60	274	106	250	8.0	9.0	0.265	0.385	15.5	13.4
39		20	19	2.36	4.60	274	106	300	8.0	9.0	0.272	0.410	16.0	13.7
40		20	19	2.36	4.60	274	106	350	8.0	9.0	0.277	0.428	16.5	13.8
41		20	19	2.36	4.60	274	106	410	8.0	9.0	0.280	0.440	17.0	14.0
42	Antoniny No. 1/3	19	18	2.08	4.16	232	91	0	17.0	14.0	0.000	0.000	17.0	14.0
43		19	18	2.08	4.16	232	91	50	17.0	14.0	0.070	0.135	18.4	15.2
44		19	18	2.08	4.16	232	91	100	17.0	14.0	0.110	0.200	19.5	16.0
45		19	18	2.08	4.16	232	91	200	17.0	14.0	0.150	0.270	20.7	17.0
46		19	18	2.08	4.16	232	91	300	17.0	14.0	0.170	0.320	21.6	17.8
47		19	18	2.08	4.16	232	91	400	17.0	14.0	0.184	0.355	22.5	18.2
48		19	18	2.08	4.16	232	91	500	17.0	14.0	0.196	0.385	23.5	18.6
49		19	18	2.08	4.16	232	91	600	17.0	14.0	0.205	0.414	24.0	19.0
50		19	18	2.08	4.16	232	91	700	17.0	14.0	0.213	0.441	24.6	19.3
51		19	18	2.08	4.16	232	91	760	17.0	14.0	0.220	0.455	25.0	19.5
52	Wonieść	70	60	3.70	7.10	500	150	0	6.5	6.6	0.000	0.000	6.5	6.6
53		70	60	3.70	7.10	500	150	30	6.5	6.6	0.500	0.250	11.0	9.0
54		70	60	3.70	7.10	500	150	60	6.5	6.6	0.660	0.450	13.5	10.8
55		70	60	3.70	7.10	500	150	90	6.5	6.6	0.800	0.600	15.4	12.0
56		70	60	3.70	7.10	500	150	120	6.5	6.6	0.900	0.740	17.0	13.2
57		70	60	3.70	7.10	500	150	180	6.5	6.6	0.990	0.860	19.0	14.8
58		70	60	3.70	7.10	500	150	240	6.5	6.6	1.060	0.970	21.4	16.0
59		70	60	3.70	7.10	500	150	300	6.5	6.6	1.120	1.070	23.0	17.6
60		70	60	3.70	7.10	500	150	360	6.5	6.6	1.175	1.160	24.0	18.5
61		70	60	3.70	7.10	500	150	450	6.5	6.6	1.220	1.240	25.0	19.2
62	Koszyce VIII A	100	95	2.00	14.00	650	130	0	10.0	13.0	0.000	0.000	10.0	13.0
63		100	95	2.00	14.00	650	130	730	10.0	13.0	0.500	2.200	26.0	20.0
64		100	95	2.00	14.00	650	130	1100	10.0	13.0	0.670	3.200	31.0	23.0
65		100	95	2.00	14.00	650	130	2100	10.0	13.0	0.770	3.900	37.0	26.0
66		100	95	2.00	14.00	650	130	3100	10.0	13.0	0.850	4.480	42.0	28.5
67		100	95	2.00	14.00	650	130	4100	10.0	13.0	0.900	4.920	45.0	30.0
68		100	95	2.00	14.00	650	130	5100	10.0	13.0	0.940	5.200	48.0	32.0
69	Koszyce X A	35	32	2.00	14.00	650	130	0	10.0	13.0	0.000	0.000	10.0	13.0
70		35	32	2.00	14.00	650	130	730	10.0	13.0	0.150	0.750	14.0	15.4
71		35	32	2.00	14.00	650	130	1100	10.0	13.0	0.200	1.020	15.8	16.3
72		35	32	2.00	14.00	650	130	2100	10.0	13.0	0.240	1.260	19.0	18.2
73		35	32	2.00	14.00	650	130	3100	10.0	13.0	0.270	1.450	21.0	19.0
74		35	32	2.00	14.00	650	130	4100	10.0	13.0	0.297	1.590	22.0	19.5
75		35	32	2.00	14.00	650	130	5100	10.0	13.0	0.320	1.700	23.0	20.0
76	Koszyce X B	65	60	2.00	14.00	650	130	0	10.0	13.0	0.000	0.000	10.0	13.0
77		65	60	2.00	14.00	650	130	730	10.0	13.0	0.280	1.320	17.0	19.0
78		65	60	2.00	14.00	650	130	1100	10.0	13.0	0.380	1.870	20.0	22.0
79		65	60	2.00	14.00	650	130	2100	10.0	13.0	0.434	2.300	25.0	27.0
80		65	60	2.00	14.00	650	130	3100	10.0	13.0	0.475	2.650	28.0	30.0
81		65	60	2.00	14.00	650	130	4100	10.0	13.0	0.510	2.870	30.0	31.5
82		65	60	2.00	14.00	650	130	5100	10.0	13.0	0.540	3.040	32.0	33.0

**Table 3 materials-16-00125-t003:** Matrix of linear simple correlation coefficients *r* for peat.

Variable	*X*1 *σp*	*X*3 *hp*	*X*5 *wp*	*X*7 *t*	*X*8 *τp*0	*Y*1 *sp*	*Y*3 *τp*
*X*1 *σp*	1.000						
*X*3 *hp*	0.169	1.000					
*X*5 *wp*	**0.779**	−0.026	1.000				
*X*7 *t*	**0.400**	**−0.272**	**0.591**	1.000			
*X*8 *τp*0	−0.187	**−0.378**	**−0.220**	0.170	1.000		
*Y*1 *sp*	**0.617**	**0.542**	**0.382**	**0.252**	**−0.288**	1.000	
*Y*3 *τp*	**0.535**	−0.165	**0.400**	**0.683**	**0.331**	**0.602**	1.000

**Table 4 materials-16-00125-t004:** Matrix of linear simple correlation coefficients *r* for gyttja.

Variable	*X*2 *σg*	*X*4 *hg*	*X*6 *wg*	*X*7 *t*	*X*9 *τg*0	*Y*2 *sg*	*Y*4 *τg*
*X*2 *σg*	**1.000**						
*X*4 *hg*	**0.757**	1.000					
*X*6 *wg*	**0.713**	**0.632**	1.000				
*X*7 *t*	**0.450**	**0.666**	**0.257**	1.000			
*X*9 *τg*0	**0.481**	**0.619**	**0.714**	**0.391**	1.000		
*Y*2 *sg*	**0.744**	**0.723**	**0.369**	**0.821**	**0.402**	1.000	
*Y*4 *τg*	**0.678**	**0.647**	**0.709**	**0.639**	**0.815**	**0.713**	1.000

**Table 5 materials-16-00125-t005:** Statistical multiple regression models.

Formula	*R* ^2^	±*SEE*	*±REmax* (%)	Formula Number
sp=−0.288+0.0077σp+0.213hp−0.00053wp+0.000072t	0.632	0.193	60	(4)
sg=0.0256+0.029σg−0.0065wg+0.00053t	0.881	0.411	60	(5)
τp=10,639+0.231σp−0.025wp+0.0041t+0.659τp0	0.699	4.750	40	(6)
τg=−0.655+0.127σg−0.746hg+0.027wg+0.0031t+1.125τg0	0.888	2.895	35	(7)

## Data Availability

Data available on request.

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
