# Peer review of "Assessment of the Undrained Shear Strength and Settlement of Organic Soils under Embankment Loading Using Artificial Neural Networks"

_materials, 2022, doi:10.3390/ma16010125_

Round 1
Reviewer 1 Report
This paper proposes an ANN method for predicting the settlement and undrained shear strength of organic soils subjected to embankment loading. The paper is comprehensible and may be relevant to Civil/Structural/Construction engineers, but it lacks significant contribution for a journal such as Materials due to the following:
1. There have been publications on ANN-based settlement embankment loading prediction. (e.g., Chik, Z., Aljanabi, Q.A., Kasa, A. et al. Tenfold cross validation artificial neural network modeling of the settlement behavior of a stone column under a highway embankment. Arab J Geosc 7, 4877–4887 (2014). https://doi.org/10.1007/s12517-013-1128-6). Therefore, the paper contributes little to existing research.
2. The authors have misunderstandings about ANN. In lines 341-344, for instance, the validation set should be used to determine the hyper-parameters (such as the number of nodes in hidden layers) of ANN, while the test set has nothing to do with "early stopping."
3. Now that ANN is the primary method of this paper, more information about ANN should be presented, such as previous studies on ANN prediction of settlement, how to determine the architecture of ANN, and a comparison of ANN's and other models' prediction accuracy, among others.
4. The unit in the final column of Table 1 should likely be "kPa" rather than "m." If so, similar errors also exist in Figures 10 (c), 11 (b), and 12 (c) (b)
Author Response
Please find enclosed answers

Reviewer 2 Report
The abstract is very incomplete. The abstract should explain the aims and scope of the work presented in the article rather than introducing the research topic. Keep the abstract concise. The first sentence should motivate your study, then explain clearly and concisely what you did and, in the end, your main result and conclusion.
The introduction to the article should be improved, and the structure of the article should not focus solely on the contributions and research in the field of materials engineering and on citing the different possibilities of optimization of the different parameters. There should be a clear reference to the development and application of the different regression models in other situations that can be assimilated into the one proposed by the authors. With this correction, the solidity of the article will be reinforced; besides predisposing the reader adequately to a better understanding of the methodology assumed, increasing the possibilities of discussion of the results and of being able to add a greater number of bibliographical references to the work.
First of all, it lacks a rigorous literature review and references to all the mentioned methods. Then, it does not go into details regarding geotechnical data and how they can be correlated in order to assess the settlement and undrained shear strength. This is fundamental for the following development: the authors simply apply regressions to fit the data without exploring possible cross-correlations among parameters. Last but not least, they state the superiority of the proposed method with respect to the other without comparing their performance on an independent set of data (different from the dataset used to fit the model). This is unacceptable in a scientific paper.
The authors should present a table containing the statistical specifications of the database used in the regression model, e.g., the maximum, minimum, average, and standard deviation of all input variables and output targets.
Lines 308 to 324: How did the author select seven parameters as the input variable? Why are the other important factors relating to the particle size distribution curve or grain shape ignored? I believe that authors should present strong evidence for selecting this set of input variables.
The authors should exactly present the quantities and variabilities of training, testing, and validation databases (if any) used in the regression model.
Table 4: The main criteria for examining the performance of a regression model for well predicting the target variable are the coefficient of correlation (r), coefficient of determination (r2), root mean squared error (RMSE), and the sum of squared error (SSE). Thus, the authors should examine all error components for their proposed model. Besides, the error components of the proposed regression model should be compared with the allowable limit of the error, which is acceptable for a regression model, as thought in the literature.
The reviewer sincerely believes that the conclusions are scarce and that, therefore, if the good results obtained in work are revised once again, together with the improvement of the introduction to the work, a more significant number of adequate and coherent conclusions will be obtained with the work presented.
Author Response
Please find enclosed answers

Round 2
Reviewer 1 Report
This work proposes an ANN method aimed at predicting the settlement and undrained shear strength of organic soils under embankment loading. The paper is understandable and might have relevance for Civil/Structural/Construction engineers, but unfortunately it lacks significant contribution for a journal like Materials.
Author Response
Please find enclosed reply to your review comments

Reviewer 2 Report
This manuscript investigates the application of the ANN approach to predict or estimate the settlement and undrained strength of soils based on the field data. To be honest, a great improvement was found concerning the coherent, fluent, and extended piece of writing. And the main innovation of this work is well conveyed. However, there remain a few issues to be addressed.
Highlighting the gytjja soil properties and behaviour is recommended, which helps to have a global reader's attention.
Line 215: Based on the author's knowledge, field or laboratory, Vane shear testing is inappropriate for clays with sand or silt laminations. How can authors justify organic soils? Explanation with citation is appreciated.
Line 286: Replace Figure 7 with Figure 9
Line 302: How can authors relate the consolidation process with peat?
Finally, I suggest the authors thoroughly check the manuscript for spelling mistakes and correct them with a native speaker fluent in technical English to improve the language and its quality.
Author Response
Please find enclosed review comments
